# Vascular Growth Factor Inhibition with Bevacizumab Improves Cardiac Electrical Alterations and Fibrosis in Experimental Acute Chagas Disease

**DOI:** 10.3390/biology12111414

**Published:** 2023-11-10

**Authors:** Lindice Mitie Nisimura, Roberto Rodrigues Ferreira, Laura Lacerda Coelho, Gabriel Melo de Oliveira, Beatriz Matheus Gonzaga, Marcelo Meuser-Batista, Joseli Lannes-Vieira, Tania Araujo-Jorge, Luciana Ribeiro Garzoni

**Affiliations:** 1Laboratory of Innovations in Therapies, Education and Bioproducts, Oswaldo Cruz Institute (LITEB-IOC/Fiocruz), Oswaldo Cruz Foundation (Fiocruz), Av. Brasil, 4365, Manguinhos, Rio de Janeiro 21040-900, Brazil; lindicem@gmail.com (L.M.N.); robertoferreira@ioc.fiocruz.br (R.R.F.); llacerdac@gmail.com (L.L.C.); biagonzaga04@hotmail.com (B.M.G.); marcelomeuser@gmail.com (M.M.-B.); taniaaj@ioc.fiocruz.br (T.A.-J.); 2Laboratory of Applied Genomics and Bioinnovations, Oswaldo Cruz Institute (LAGABI-IOC/Fiocruz), Rio de Janeiro 21040-900, Brazil; 3Laboratory of Cell Biology, Oswaldo Cruz Institute (LBC-IOC/Fiocruz), Rio de Janeiro 21040-900, Brazil; oliveira.gabriel.melo@gmail.com; 4Laboratory of Biology of the Interactions, Oswaldo Cruz Institute (LBI-IOC/Fiocruz), Oswaldo Cruz Foundation, Rio de Janeiro 21040-900, Brazil; lannes@ioc.fiocruz.br

**Keywords:** angiogenesis, vascular endothelial growth factor, cardiac remodeling, Chagas disease

## Abstract

**Simple Summary:**

Chagas disease is an infectious condition caused by *Trypanosoma cruzi* that especially affects the heart; the infected patient may present alterations in heartbeat and an increase in heart volume. When the infection is not treated and remains active for a long time, it can lead to heart failure and death. In many cases, microscopic heart analysis reveals many defense cells that characterize inflammation and several scars in the organ—called fibrosis. Multiple components are involved in inflammation and fibrosis development; for example, an increased blood vessel number contributes to the process as it allows for a greater quantity of cell arrival. VEFG-A is a potent vessel formation inducer. In this context, we investigated the effect of inhibition of VEGF in *T. cruzi*-infected mice, and we found that the VEGF blockage significantly increased survival, reduced inflammation, improved cardiac electrical function, diminished the vessel formation and reduced cardiac fibrosis. This work shows that VEGF is involved in cardiac alterations observed in Chagas disease and the inhibition of this factor could be a potential treatment for *T. cruzi*-infected patients.

**Abstract:**

Chagas disease (CD) caused by *Trypanosoma cruzi* is a neglected illness and a major reason for cardiomyopathy in endemic areas. The existing therapy generally involves trypanocidal agents and therapies that control cardiac alterations. However, there is no treatment for the progressive cardiac remodeling that is characterized by inflammation, microvasculopathy and extensive fibrosis. Thus, the search for new therapeutic strategies aiming to inhibit the progression of cardiac injury and failure is necessary. Vascular Endothelial Growth Factor A (VEGF-A) is the most potent regulator of vasculogenesis and angiogenesis and has been implicated in inducing exacerbated angiogenesis and fibrosis in chronic inflammatory diseases. Since cardiac microvasculopathy in CD is also characterized by exacerbated angiogenesis, we investigated the effect of inhibition of the VEGF signaling pathway using a monoclonal antibody (bevacizumab) on cardiac remodeling and function. Swiss Webster mice were infected with Y strain, and cardiac morphological and molecular analyses were performed. We found that bevacizumab significantly increased survival, reduced inflammation, improved cardiac electrical function, diminished angiogenesis, decreased myofibroblasts in cardiac tissue and restored collagen levels. This work shows that VEGF is involved in cardiac microvasculopathy and fibrosis in CD and the inhibition of this factor could be a potential therapeutic strategy for CD.

## 1. Introduction

Chagas disease (CD) is caused by the *Trypanosoma cruzi* parasite; it is endemic in Latin America and affects 7 million people worldwide [1]. The acute phase of CD is characterized by high parasitemia and unspecific symptoms, including fever and sickness. Cardiomyopathy and meningoencephalitis can be observed in more severe acute cases [1]. The immunological control of infection results in evolution to the chronic phase of CD; this stage can be asymptomatic (indeterminate) or symptomatic, which represents 30% of patients with cardiomyopathy being the major clinical manifestation [1]. CD is a neglected disease and little investment is made by the industry and governmental agencies to develop more effective therapies. Benznidazole and Nifurtimox are trypanocidal compounds that are highly effective during the acute phase of CD with an 80% cure rate. However, during the chronic stage, only 20% of treated patients are cured [2]. Moreover, the difficulties of managing adverse effects in remote areas, where access to health systems is difficult, can result in abandonment of treatment. The management of chagasic cardiomyopathy primarily involves diuretics, aldosterone antagonists, angiotensin-converting enzyme (ACE) inhibitors and antiarrhythmic agents (beta adrenergic blockers or amiodarone), which improve patients’ quality of life [3]. However, these treatments cannot reverse the cardiac remodeling caused by microvasculopathy, chronic inflammation, progressive fibrosis and hypertrophy. 

In CD patients’ hearts, vascular constrictions, microaneurysms, dilatation and occlusive thrombi are consequences of vasoactive substances such as endothelin-1 and thromboxane [4,5,6]. Cardiac exacerbated angiogenesis occurs both in the experimental *T. cruzi* infection in mice [7,8] and in chronic chagasic cardiomyopathy in humans [9]. However, the role of microvascular growth in chagasic cardiomyopathy is unknown. 

Angiogenesis is a multistep process involving vascular instability that is caused by the detachment of α-smooth muscle actin (α-SMA)-positive mural cells from the vascular wall [10,11]. In inflammatory disorders, angiogenesis is exacerbated and abnormal, contributing to tissue remodeling and pathogenesis, because of mural cell differentiation to myofibroblasts; these changes generate and maintain chronic fibrotic processes [12,13]. This hypothesis is also supported by studies of fibrogenesis in the liver [14,15] and kidneys [16] and showed that mural cells are a source of myofibroblast and strongly strengthen the role of angiogenesis in fibrosis [15,17].

Vascular endothelial growth factor-A (VEGF-A) is a powerful polypeptide regulator of blood vessel function [18,19]. It is a key proangiogenic factor and inducer of vascular hyperpermeability that contributes to inflammatory cell migration [20]. VEGF-A activates the receptor tyrosine kinase VEGFR-2/Flk-1, inducing extracellular signal-regulated protein kinase (ERK) 1/2 phosphorylation, cellular proliferation and vascular growth [19,21].

Bevacizumab is a VEGF inhibitor and antiangiogenic agent that is clinically used to treat some malignant tumors and chronic inflammatory diseases [22,23]. VEGF inhibition has also shown anti-fibrotic effects in experimental models of urethral, hepatic, articular and spinal epidural fibrosis [24,25,26,27]. In this context, we hypothesized that the inhibition of VEGF using bevacizumab could help to alleviate chagasic cardiomyopathy. Here, we tested the effect of VEGF inhibition with bevacizumab in an experimental model of acute CD in mice. Our results showed that the *T. cruzi* infection increased cardiac VEGF expression and that bevacizumab treatment reduced cardiac angiogenesis and fibrosis, improved cardiac electrical activity and increased survival in treated *T. cruzi*-infected animals.

## 2. Materials and Methods

### 2.1. Infection of Animals with Trypomastigote Forms of T. cruzi

Male Swiss Webster mice (age 6–8 weeks; weight 18–20 g) bred in-house in the Institute of Science and Technology in Biomodels (ICTB) from Oswaldo Cruz Foundation were infected intraperitoneally with 10^4^ blood forms trypomastigotes of *T. cruzi* (Y strain) that were maintained through Swiss Webster mice infections. The animals were separated into the following groups: non-infected and not-treated (NI NT), non-infected and treated with anti-VEGF (NI T), infected and not-treated (Y NT) and infected and treated with anti-VEGF (Y T) and euthanized on day 8 and 15 post-infection (dpi). 

### 2.2. Anti-VEGF Antibody Treatment

Treated mice received 5 mg/kg of Avastin^®^ (bevacizumab from Roche, Basel, Switzerland) intraperitoneally (i.p.) (previously described as the dose having an antiangiogenic effect [28,29,30]) 3 times a week starting at 3 dpi until reaching 15 dpi. The control group received the vehicle phosphate-buffered saline (PBS) [0.01 M phosphate buffer, 0.0027 M potassium chloride and 0.137 M sodium chloride].

### 2.3. Parasitological Parameters

Parasitemia was performed with Pizzi–Brener method [31] in which 5 µL of fresh blood taken from a mouse’s tail were placed on a glass slide, covered with an 18 × 18 mm glass coverslip, and the parasite’s number was quantified in 50 randomly observed microscopical fields, covering the entire area of the coverslip under overall magnification of 400. The number of parasites/mL was obtained by multiplying the number by the correction factor, which was 0.72 × 10^4^ in our microscope. Mortality and body weight were monitored regularly for 15 days. 

### 2.4. Electrocardiographic Studies

Mice were i.p. tranquilized with diazepan (20 mg/kg) and transducers were carefully placed under the skin in accordance with chosen preferential derivation (DII). Traces were recorded using a digital system (Power Lab 2/20) connected to a bio-amplifier in 2 mV for 1 s (AD Instruments Company, Sydney, Australia). Filters were standardized between 0.1 and 100 Hz and traces were analyzed using the LabChart for Windows v8.1.23 Software (AD Instruments Company). We continuously measured for the automatic traces using LabChart Reader v8.1.22 software for 30 min (AD Instruments Company). The parameters evaluated by the software comprised heart rate (beats per minute—bpm), duration of the PR, QRS, QT intervals and P wave in ms (millisecond) at 14 dpi. The relationship between the QT interval and RR interval was individually assessed. However, due to the physiologically accelerated heart rate, QT interval was corrected (QTc) with Hodges formula [32]. We qualified and classified the possible arrhythmia types by evaluating each murine trace. In addition, the LabChart Reader v8.1.22 showed irregular (time) intervals during the murine ECG. This indicated arrhythmia in the trace.

### 2.5. Histopathology and Immunofluorescence 

Half of the heart was processed for microscopy analysis in optimal cutting temperature compound (OCT), sectioned (5 µm thick) in Leica -CM1850 cryostat at −22 °C and collected in poly-L-lysine-coated glass slides. The sections were stained with sirius red (Sigma–Aldrich, St. Louis, MI, USA) or with hematoxylin/eosin (EasyPath, Indaiatuba, Brazil) and examined using light microscopy, especially focusing on the left ventricle. A total of 3–5 animals samples were analyzed for each experimental group in Zeiss Axioplan 2 (Carl Zeiss, Jena. Germany) microscope at an overall magnification of 400.

The software ImageJ (National Institutes of Health, USA available at https://imagej.nih.gov/ij/index.html, accessed on 10 July 2023) was used for the quantification of tissue parasitism, inflammation and collagen area as a percentage of the total area as described by Grishagin, 2015 [33]. The analyses were carried out in at least 15 fields per slice from 5 animals per group.

For immunofluorescence, heart sections were stained with specific primary mouse anti-α-SMA (Sigma–Aldrich, USA) antibody 5 μg/mL overnight at 4 °C and secondary goat anti-mouse Alexa Fluor 594 antibody at 1 μg/mL (Thermo Fisher Scientific, Waltham, MA, USA) was incubated for 1 h at room temperature. FITC-conjugated *Griffonia simplicifolia* I lectin at 50 μg/mL was used to stain endothelial cells for 30 min at room temperature and 4′,6-diamidino-2-phenylindole [(DAPI) Thermo Fisher Scientific, Waltham, MA, USA] at 0.2 μg/mL was used for DNA visualization. Slides were examined with the Zeiss Axioplan 2 microscope equipped with epifluorescence. Further image processing was performed with Adobe Photoshop software version 13.0 x32 (Adobe Systems Inc., San Jose, CA, USA). 

### 2.6. Immunoblotting

Ventricular heart proteins from each group were extracted from 100 mg tissue/mL of phosphate-buffered saline to which 0.4 M sodium chloride, 0.05% Tween 20 and protease inhibitors [0.1 mM phenylmethylsulfonyl fluoride (Sigma Aldrich, USA) and protease (Roche, Indianapolis, IN, USA) and phosphatase inhibitors cocktail (Sigma Aldrich, USA)] were added. The samples were sonicated twice and centrifuged for 10 min at 3000× *g*, and the supernatant was stored at −80 °C. Total proteins in the lysates (20–40 µg/lane) were separated by SDS-PAGE (10%) and transferred to nitrocellulose membranes (Bio-Rad, Hercules, CA, USA). Non-specific binding sites were blocked by incubating the membranes with 5% (*w*/*v*) nonfat milk/TBS/0.1% Tween-20 for 1 h at room temperature. 

The membranes were probed with the specific primary antibody rabbit anti-VEGF (42 kDa) (ABCAM, Cambridge, UK) at 1 μg/mL overnight at 4 °C. For a loading control, a mouse anti-Glyceraldehyde 3-phosphate dehydrogenase (GAPDH, 36 kDa) monoclonal antibody from Fitzgerald (Acton, MA, USA) was incubated for 1 h at room temperature. The membranes were incubated with secondary goat anti-rabbit IgG or goat anti-mouse IgG HRP-labeled antibody (Thermo Fisher Scientific, Waltham, MA, USA) at 0.08 μg/mL for 1 h at room temperature, followed by treatment using the Supersignal chemiluminescent kit (Pierce, Rockford, IL, USA) and exposition by X-ray film. Densitometric analysis of the bands was performed with the Image Studio Lite version 4.0 software.

### 2.7. Ethics Statement

The use of animals and experimental procedures was in accordance with Brazilian Law 11.794/2008 and regulations of the National Council of Animal Experimentation Control (CONCEA). The mice were housed at a maximum of 6 individuals per cage, kept in a specific pathogen-free (SPF) room at 20 to 24 °C under a 12 light and 12 h dark cycle and were provided sterilized water and chow ad libitum. All experimental animal procedures were performed following the license (LW—40/13) approved by the Ethics Committee for Animal Use the Oswaldo Cruz Foundation (CEUA/FIOCRUZ). All animals were euthanized using an anesthetics overdose followed by cervical dislocation on the 8th and 15th day after experimental infection was carried out with *T. cruzi*.

## 3. Results 

### 3.1. Anti-VEGF Treatment Improves Survival Rate and Inflammation

We treated the infected mice with anti-VEGF (bevacizumab) i.p. 3 times a week starting at 3 dpi; evaluation of parasitemia revealed typical *T. cruzi* trypomastigote peak at 8 dpi (Figure 1A). *T. cruzi-*infected mice exhibited weight loss beginning at 10 dpi (Figure 1B). Anti-VEGF administration significantly increased the survival rate, which was 60% in the treated mice compared to 25% in the not-treated and infected group (Figure 1C), and reduced parasitemia at 8 dpi (Figure 1A) but no differences in weight were observed (Figure 1B). 

Cardiac parasitism and inflammation were evaluated at 8 and 15 dpi (Figure 2A–D). The peak of tissue parasitism and inflammation occurred at 15 dpi (Figure 2E,F). At 8 dpi, we observed a lower percentage of inflammation in the tissues from infected and anti-VEGF-treated mice (Figure 2F). At 15 dpi, in addition to the decrease in inflammation, a significant reduction in amastigote number was observed in the myocardium of infected and anti-VEGF-treated mice (Figure 2E,F). 

### 3.2. VEGF Protein Expression in Cardiac Tissue

Next, we determined the expression of VEGF in cardiac tissue with immunoblotting (Figure 3). At 15 dpi, the infected and non-treated group presented a 2.2-fold increase in cardiac VEGF expression when compared to non-infected controls (*p* < 0.01), and we did not observe changes in VEGF levels at 8 dpi when compared to non-infected controls (Figure 3A,B). The treatment with bevacizumab did not change VEGF levels at 8 dpi; however, at 15 dpi, treatment with bevacizumab induced a 3.6-fold reduction in VEGF expression compared to the infected non-treated group (*p* < 0.001) (Figure 3C,D). 

### 3.3. Anti-VEGF Treatment Reduces the Blood Vessel Abundance in Cardiac Tissue

We further investigated the blood vessel abundance in hearts under normal and *T. cruzi-*infected conditions at 8 and 15 dpi. Heart tissues were stained with FITC-conjugated *Griffonia simplicifolia* I lectin to determine the number of blood vessels and with antibody against α-smooth muscle actin (α-SMA) to evaluate vascular maturity in non-infected and non-treated (Figure 4A), non-infected and treated (Figure 4B), *T. cruzi*-infected and non-treated (Figure 4C), *T. cruzi*-infected and treated samples at 8 dpi (Figure 4D), *T. cruzi*-infected and non-treated (Figure 4E) and *T. cruzi*-infected and treated samples at 15 dpi (Figure 4F). Quantification analysis showed a significant increase in areas stained with lectin (Figure 4G) and α-SMA (Figure 4H) in cardiac tissue of the *T. cruzi-*infected and non-treated group at 15 dpi (34.35% and 13%, respectively) compared to the non-infected group (19.7% and 0.36%, respectively). We observed that the treatment with bevacizumab reduced (23%) the lectin-stained area (green in Figure 4E,F), showing a reduction in the number of blood vessels. The treatment did not change the α-SMA-stained area. 

### 3.4. Treatment Prevents Heart Fibrosis Development in T. cruzi-Infected Mice

We evaluated heart fibrosis by picrosirius red staining of heart sections. After infection, we observed a progressive increase in collagen deposition, which was visualized in red in sections from infected and not-treated mice (Figure 5C,E) compared to hearts from non-infected mice (Figure 5A,B). Treatment with anti-VEGF prevented this collagen deposition (Figure 5D,F). Quantification analysis revealed a reduction in the percentage of areas with collagen in infected and treated samples (1.15%) compared to infected and not-treated samples (35.9%) at 15 dpi (Figure 5G).

### 3.5. VEGF Blockage Ameliorates Electrocardiographic Alterations Caused by T. cruzi 

The cardiac electrical conduction system was evaluated using ECG analysis (Figure 6). ECG demonstrated increased PR interval time in *T. cruzi*-infected animals (57.0 ms) compared to the non-infected control group (28.5 ms), and anti-VEGF treatment minimized the PR time conduction in the experimental model (45.3 ms) (Figure 6B). Furthermore, the heart rate was reduced in infected and non-treated mice (341.4 bpm) compared to NI controls (765 bpm), and treatment significantly improved the heart rate (448.1 bpm) (Figure 6E). QTc interval times (NI NT: 26.2 and NI T: 26.9 ms) were decreased in infected and not-treated mice (24.4 ms) and infected and treated mice (15.9 ms) compared to non-infected mice (NI NT: 26.2 ms) (Figure 6D). No difference was observed in the QRS parameter between the groups (Figure 6C).

Cardiac dysfunction, such as atrioventricular block (AVB, dashed circle), sinusal arrhythmia (ART, dashed line) and sinusal bradycardia (BRD, black line), was detected in *T. cruzi*-infected mice (Figure 7C,D). Anti-VEGF treatment decreased the incidence and severity of BRD in the treated groups (Figure 7E). 

## 4. Discussion

In this study, we investigated the effect of the VEGF inhibition with bevacizumab, an antiangiogenic monoclonal antibody, in an experimental murine model of acute CD. The experimental model that was used is an important model for the study of the acute Chagas disease that gives a rapid assessment of the experimental acute infection kinetics and enables to conduct pathogenicity and chemotherapy (among others) investigations in this field [34,35,36,37]. We were the first to show that VEGF blockage significantly increased the survival of mice, reduced VEGF levels, decreased angiogenesis, and inhibited inflammation, fibrosis and alterations in cardiac electrical function. Vascular injury is one of the first alterations observed in chagasic myocardiopathy [38,39]. Vascular injuries in the hearts of patients with CD such as vascular constrictions, microaneurysms, dilatation and occlusive thrombi are consequences of vasoactive substances such as endothelin-1 and thromboxane [4,5,6]. 

In general, angiogenesis occurs in response to ischemia, improving the oxygen and nutrient delivery [26]; however, angiogenesis is also involved in fibrosis in some inflammatory disorders [15,17], and, in CD, the inflammatory process contributes to cardiac remodeling [40,41]. Furthermore, we also described cerebral vasculopathy in mice during acute CD which was characterized by obstructive plugging, microvascular inflammation and functional capillary rarefaction [42], corroborating with other data in the literature that demonstrated vascular damage and microvasculopathy in Chagas disease [43,44,45].

Once we identified increased VEGF protein expression in cardiac tissue of Swiss Webster mice during acute infection, we decided to investigate whether the modulation of this factor would have a beneficial effect on the development of experimental Chagas disease and cardiac alterations. VEGF-A is a proangiogenic factor and induces vascular hyperpermeability that could contribute to inflammation [18]. Previous studies showed that the expression and production of cardiac angiogenic mediators depend on the *T. cruzi* genetic population. The authors observed that in the experimental C57BL/6 mice infection, cardiac angiogenic mediators are increased during *T. cruzi* Y strain infection. Elevated expression of cardiac VEGF, Ang-1 and Ang-2, and reduced production of TNF and CCL5 were observed in the Y strain infection compared to the infection with the Colombian strain [8]. 

Our findings show that the inhibition of the VEGF pathway by bevacizumab-modulated angiogenesis, inflammation and fibrosis were also described in other tissues in chronic inflammatory conditions, such as during the treatment of rheumatoid arthritis [46]. Moreover, we show that the VEGF neutralization with bevacizumab reduced the VEGF levels and the parasitism in infected animals, suggesting a direct effect of VEGF on the parasite; however, further studies need to be conducted. 

Another molecule that plays an important role in Chagas’ cardiopathy and in the life cycle of the parasite is TGF-β [40,45,47]. Interestingly, our group have been studying the role of TGF-β on the *T. cruzi* infection and Chagas disease development. We demonstrated that the TGF-β inhibition also impairs the *T. cruzi* infection in cardiomyocytes and parasite cycle completion [48]. Although we did not evaluate the effect of TGF-β modulation on the development of cardiac angiogenesis during Chagas disease, inhibition of its signaling pathway reduces fibrosis and improves cardiac function [40,45,47], as observed in the present study by inhibition of the VEGF pathway.

To verify the effect of the *T. cruzi* infection and the effect of bevacizumab treatment on angiogenesis, we stained the cardiac tissue with FITC-conjugated *Griffonia simplicifolia*. This lectin binds specifically to endothelial cells, allowing evaluation of the vascular bed [49]. Concomitantly, we stained α-smooth muscle actin-positive cells to verify the distribution of myofibroblasts in cardiac tissue. Mural cells that detach from the vascular wall during the angiogenic process could differentiate in myofibroblasts and contribute to fibrosis [12,13]. Studies of fibrogenesis in the liver [15] and kidneys [16] showed that mural cells are a source of myofibroblast progenitors that contribute to fibrosis and tissue remodeling [15,17].

We observed a significant increase in the vascular and myofibroblast-stained areas in *T. cruzi*-infected heart tissue. Bevacizumab treatment significantly reduced the vascular area and resulted in a decreased trend of α-smooth muscle actin-positive cells. Next, we evaluated the collagen levels in cardiac tissue by sirius red staining, which is the standard method to evaluate the organization of collagen fibers in tissues [50]. We found an increase in the stained areas of infected animals and a significant reduction in cardiac fibrosis after bevacizumab treatment. Huang et al. (2013) evaluated the effects of bevacizumab on the formation of fibrosis and also described that this antibody could alleviate liver fibrosis by blocking the effect of VEGF on hepatic stellate cells [25]. Another study demonstrated the effect of bevacizumab on peritoneal fibrosis in a rat model. Regarding the histopathological findings, fibrosis also significantly decreased in bevacizumab groups [51]. 

Finally, we evaluated the ECG murine traces. Compromise in the cardiac conduction system during the experimental *T. cruzi* infection caused an inflammatory response in the humoral and cellular immune system and others. Consequently, it also promotes changes in the cardiac conduction system and extracellular matrix remodeling. We observed an increase in the PR interval and cardiac arrhythmia presence, demonstrated by synusal bradycardia. QTc time decrease was not relevant to the ECG. The bevacizumab treatment inhibited these electrical alterations in the heart, suggesting a cardiac protective effect of this VEGF-neutralizing antibody in infected animals. 

Contradictory to our data, it was described that a higher cardiovascular risk factor is present in bevacizumab-treated cancer patients [52,53]. However, the mechanisms are not fully understood but may be the result of exacerbated inhibition of VEGF, decreasing nitric oxide NO and/or increasing proinflammatory gene expression which induces vasoconstriction and platelet aggregation. Nevertheless, these different effects could be explained as they were correlated to the bevacizumab dose [52,53] and thus close monitoring of treatment should be carried out. 

Considering that a major problem with Chagas disease is chronic cardiomyopathy, in which there is well-established fibrosis, the VEGF blockage could be beneficial to minimize this aspect; however, further studies are necessary to understand the VEGF role at this stage. We recognize the high cost of the antibody and the consequent difficulty of its broad use as a treatment for Chagas disease; however, once the participation of VEGF is demonstrated, we hope that it will provoke the investigation of low-cost and accessible inhibitors.

## 5. Conclusions

In conclusion, this study shows that VEGF is involved in cardiac collagen deposition and cardiac conduction system dysfunction during *T. cruzi* infection. Altogether, our data suggested that the therapeutic effects of VEGF inhibition could be a potential strategy for the inhibition of the progression of chagasic cardiomyopathy. 

## Figures and Tables

**Figure 1 biology-12-01414-f001:**
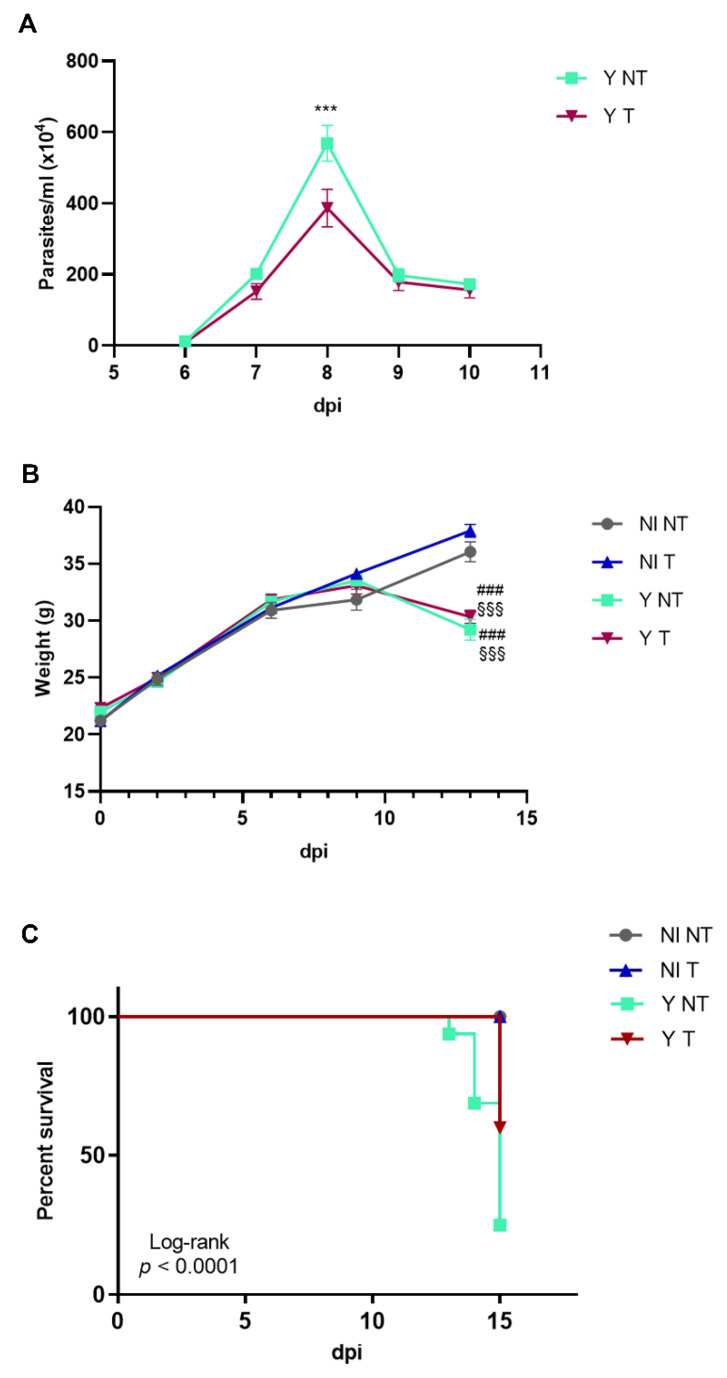
Anti-VEGF treatment of Swiss Webster mice infected with the Y strain of *T. cruzi.* Mice were intraperitoneally infected with 10^4^ blood trypomastigote forms and anti-VEGF antibody (5 mg/kg) or PBS buffer was administered intraperitoneally at 3 dpi, 3 times a week for 15 days. The following parameters were evaluated in a kinetic study: (**A**) parasitemia, (**B**) body weight and (**C**) survival rate in non-infected and not-treated (NI NT) mice, non-infected mice treated with anti-VEGF (NI T), and infected and not-treated (Y NT) and treated (Y T) mice. (**A**) The parasitemia peak occurred at 8 dpi. (**B**) Body weight loss started at 10 dpi in infected mice. (**C**) At 15 dpi, 60% of infected and treated mice survived compared to 25% of the infected animals. Mean ± SEM. Two-way ANOVA test (parasitemia and weight) or Log-rank test (survival). ### *p* < 0.001 versus NI NT; §§§ *p* < 0.001 versus NI T; *** *p* < 0.001 versus Y T. *n* = 10 animals/group.

**Figure 2 biology-12-01414-f002:**
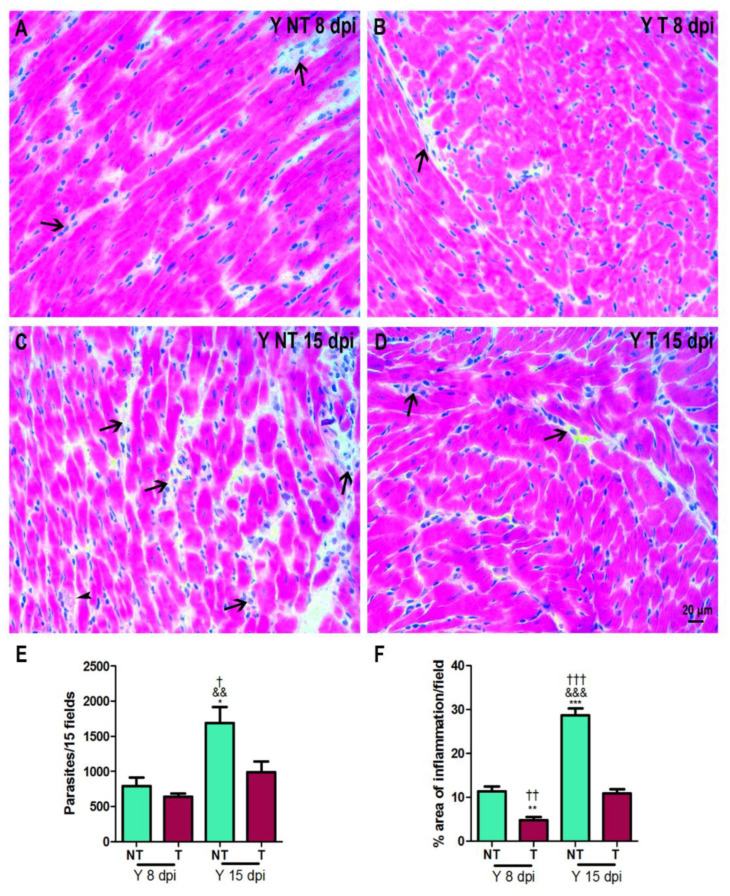
Heart histopathology. Hematoxylin/eosin staining of heart samples from intraperitoneally infected mice with 10^4^ Y blood trypomastigote forms and anti-VEGF antibody (5 mg/kg) or PBS buffer was administered intraperitoneally at 3 dpi, 3 times a week. (**A**) Infected and not-treated (Y NT) mice at 8 dpi; (**B**) infected and treated with anti-VEGF (Y T) mice at 8 dpi; (**C**) infected and not-treated (Y NT) mice at 15 dpi; and (**D**) infected and treated (Y T) mice at 15 dpi. (**E**) Quantification of the number of parasites (arrowhead). (**F**) Evaluation of the percentage of inflammation area (arrow). The peak of tissue parasitism and inflammation can be observed in infected and non-treated mice at 15 dpi. At this time, anti-VEGF treatment reduced both parameters. Bar = 20 µm. Mean ± SEM. One-way ANOVA test, † *p* < 0.05; †† *p* < 0.01; ††† *p* < 0.001 versus Y NT at 8 dpi; && *p* < 0.01; &&& *p* < 0.001 versus Y T at 8 dpi; * *p* < 0.05; ** *p* < 0.01; *** *p* < 0.001 versus Y T at 15 dpi. *n* = 5 animals/group.

**Figure 3 biology-12-01414-f003:**
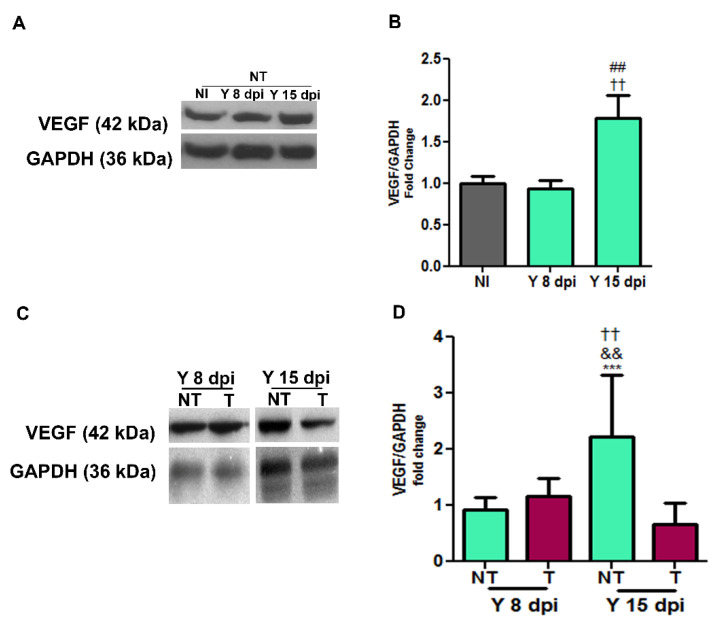
Western blotting of cardiac VEGF from mice intraperitoneally infected with 10^4^ Y strain blood trypomastigote forms and anti-VEGF antibody (5 mg/kg) or PBS buffer was administered intraperitoneally at 3 dpi, 3 times a week for 8 or 15 days. Representative immunoblotting of VEGF (42 kDa) expression at 8 and 15 dpi (**A**,**B**). Cardiac tissues were harvested, and protein lysates were probed with an anti-VEGF antibody. GAPDH (36 kDa) was used as a loading control. It is possible to observe expression levels of VEGF in the following distinct experimental groups: non-infected and non-treated (NI NT), non-infected and treated (NI T), infected and non-treated (YN NT) and infected and treated (Y T). Notice the increase in VEGF expression levels in infected animals compared to non-infected controls and the reduction in VEGF levels induced with bevacizumab (**C**,**D**). Values are expressed as fold change. ## *p* < 0.01 versus NI; †† *p* < 0.01 versus Y NT at 8 dpi; && *p* < 0.01 versus Y T at 8 dpi; *** *p* < 0.001 versus Y T at 15 dpi. One-way ANOVA test. *n* = 5–10 animals/group.

**Figure 4 biology-12-01414-f004:**
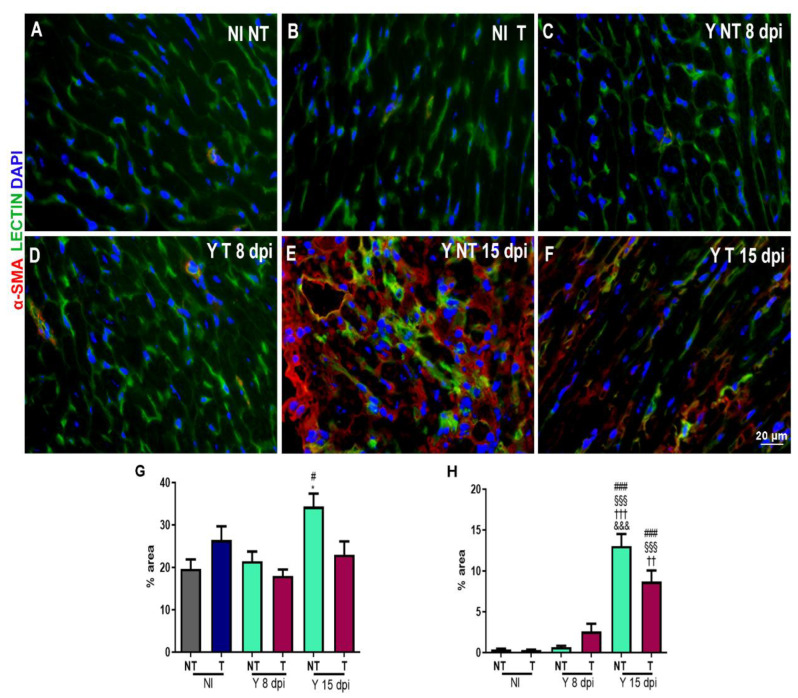
Immunofluorescence of α-SMA and vessels in cardiac tissue from mice intraperitoneally infected with 10^4^ Y blood trypomastigote forms and anti-VEGF antibody (5 mg/kg) or PBS buffer was administered intraperitoneally at 3 dpi, 3 times a week for 8 or 15 days. Immunofluorescence of cardiac vessels (green), α-SMA (red) and DNA (blue) of (**A**) non-infected and not-treated (NI NT); (**B**) non-infected and treated (NI T); (**C**) and infected and not-treated (Y NT) samples at 8 dpi; (**D**) infected and treated (Y T) samples after 8 days; (**E**) infected and not-treated (Y NT) samples at 15 dpi; and (**F**) infected and treated (Y T) samples after 15 days. (**E**,**F**) After 15 days of infection, an increase in cardiac vessels and α-SMA staining was observed, and the treatment minimized these alterations. Bar graph shows the quantification of the percentage of the stained areas of (**G**) lectin and (**H**) α-SMA. An increase in lectin and α-SMA levels was observed in *T. cruzi*-infected animals after 15 days. Anti-VEGF treatment reduced the lectin-stained area. Bar = 20 µm. Mean ± SEM. One-way ANOVA test, # *p* < 0.05; ### *p* < 0.001 versus NI NT; §§§ *p* < 0.001 versus NI T; †† *p* < 0.01; ††† *p* < 0.001 versus Y NT at 8 dpi; &&& *p* < 0.001 versus Y T at 8 dpi; * *p* < 0.05 versus Y T at 15 dpi. *n* = 3 animals/group.

**Figure 5 biology-12-01414-f005:**
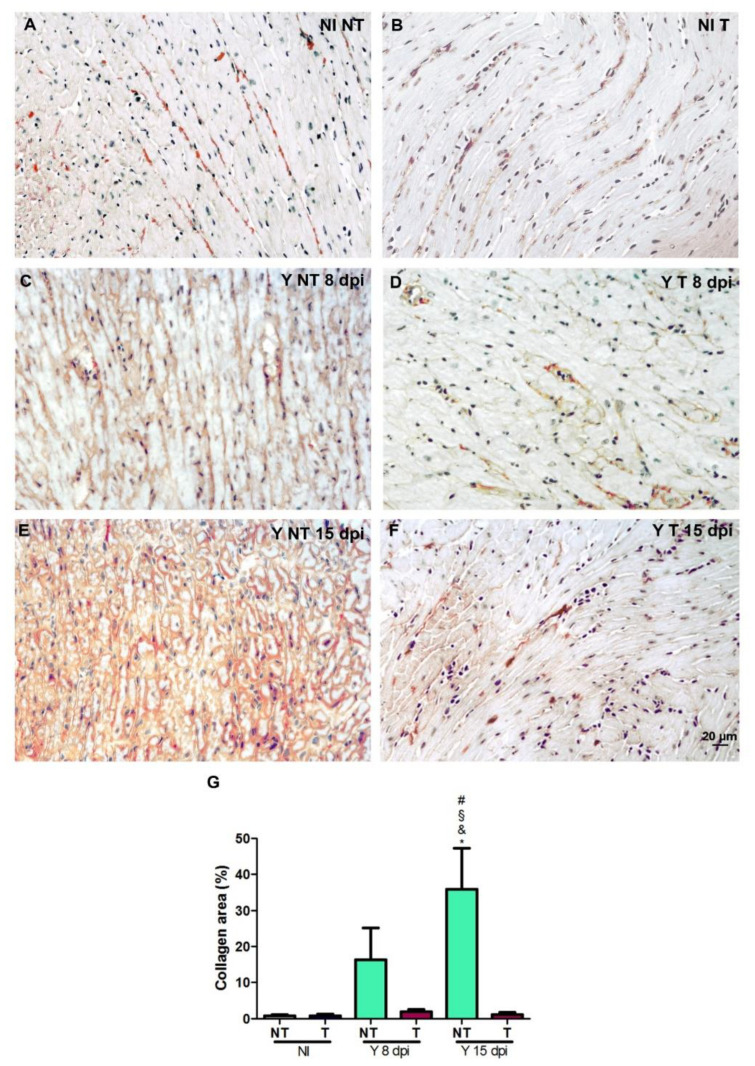
Histochemical analysis of fibrosis in heart from mice intraperitoneally infected with 10^4^ Y strain blood trypomastigote forms and anti-VEGF antibody (5 mg/kg) or PBS buffer was administered intraperitoneally at 3 dpi, 3 times a week for 8 or 15 days. Picrosirius red staining shows the collagen distribution in (**A**) non-infected and not-treated (NI NT); (**B**) non-infected and treated (NI T); (**C**) infected and not-treated (Y NT) at 8 dpi; (**D**), infected and treated (Y T) at 8 dpi; (**E**) infected and not-treated (Y NT) at 15 dpi; and (**F**) infected and treated (Y T) at 15 dpi samples. (**G**) Quantification of the percentage of collagen area. A progressive increase in collagen staining is observed in *T. cruzi*-infected and not-treated hearts, and anti-VEGF treatment reduced the percentage of the collagen area. Bar = 20 µm. Mean ± SEM. One-way ANOVA test, # *p* < 0.05 versus NI NT; § *p* < 0.05 versus NI T; & *p* < 0.05 versus Y T at 8 dpi; * *p* < 0.05 versus Y T at 15 dpi. *n* = 5 animals/group.

**Figure 6 biology-12-01414-f006:**
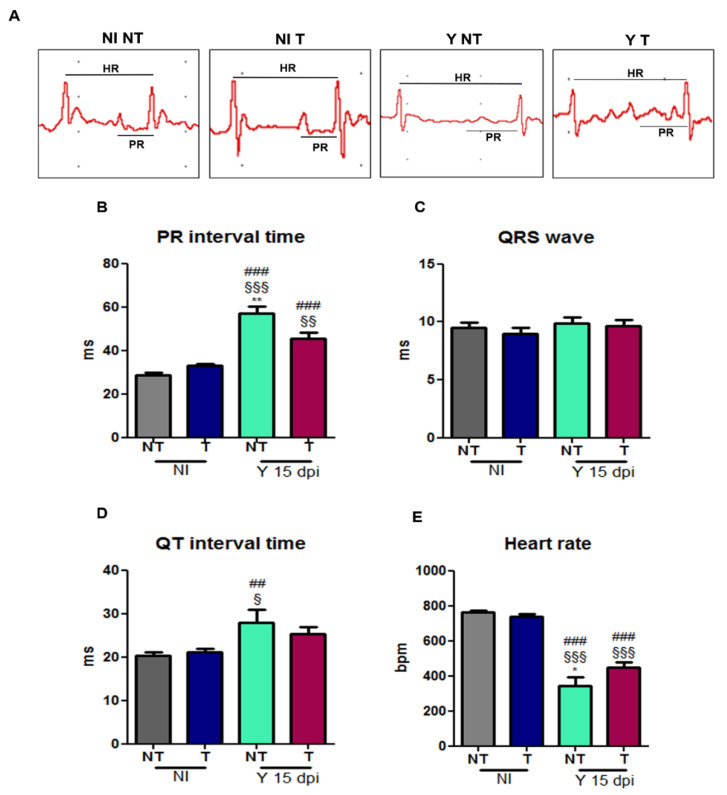
Electrocardiographic alterations in mice intraperitoneally infected with 10^4^ Y blood trypomastigote forms and anti-VEGF antibody (5 mg/kg) or PBS buffer treatment administered intraperitoneally at 3 dpi, 3 times a week for 15 days. (**A**) Representative electrocardiographic traces of groups at 15 dpi. The cardiac electric conduction system was calculated. ECG parameters were evaluated in time conduction (millisecond) during experimental acute CD. (**B**) PR interval, (**C**) QRS wave, (**D**) QTc interval and (**E**) HR were evaluated in non-infected and not-treated (NI NT), non-infected and treated (NI T), infected and not-treated (Y NT) and infected and treated (Y T) mice. Mean ± SEM. One-way ANOVA test, ## *p* < 0.01; ### *p* < 0.001 versus NI NT; § *p* < 0.05 §§ *p* < 0.01; §§§ *p* < 0.001 versus NI T; * *p* < 0.05; ** *p* < 0.01 versus Y T at 15 dpi. *n* = 10 animals/group.

**Figure 7 biology-12-01414-f007:**
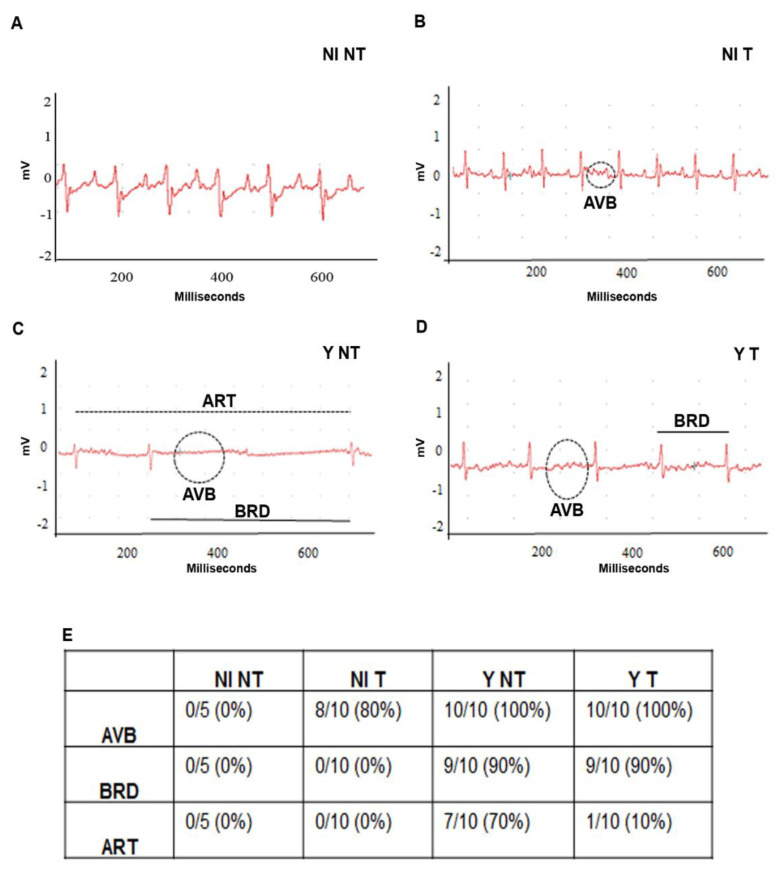
Conduction disorders in acute CD and anti-VEGF antibody (5 mg/kg) of intraperitoneally administered treatment at 3 dpi, 3 times a week for 15 days. Electrical conduction disturbances are highlighted in the representative electrocardiographic tracings at 15 dpi. (**A**) non-infected and not-treated (NI NT), (**B**) non-infected and treated (NI T), (**C**), infected and not-treated (Y NT) and (**D**) infected and treated (Y T) groups. (**E**) The number of electrical conduction disturbances in the groups. The compromise of cardiac electrical conduction was promoted by *T. cruzi* during the acute experimental phase. The treatment promotes better incidence of sinus bradycardia in infected mice *n* = 5–10 animals/group.

## Data Availability

Data are contained within the article and Appendix A.

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
