# Peer review of "Vascular Growth Factor Inhibition with Bevacizumab Improves Cardiac Electrical Alterations and Fibrosis in Experimental Acute Chagas Disease"

_biology, 2023, doi:10.3390/biology12111414_

Round 1
Reviewer 1 Report
Overall, this is a well written manuscript identifying a role for VEGF in pathogenesis of cardiac inflammation and fibrosis in a mouse model of acute Trypanosoma cruzi infection, as a model of Chagas disease. The results have implications for developing novel therapies to improve treatment of Chagas disease. A few minor items should be clarified and the discussion should include implications for such a treatment in chronic disease.
1. In the materials and methods, please list the source of the Male Swiss Webster mice. Were they bred in house or purchased from a commercial vendor?
2. What was the source of the T. cruzi Y parasites used for infection? Were they maintained in mammalian cell culture or obtained as blood form trypomastigotes from infected mice?
3. In line 111, please remove “de”
4. In section 2.1 or 2.2, please specify how many days after infection and/or treatment mice were euthanized for histologic analysis.
5. In line 121, please list the type of microscope and magnification used to analyze sections of heart stained with picrosirius red and hematoxylin/eosin
6. In line 304, please capitalize Chagas disease
7. The data shown is very convincing that inhibition of VEGF in an experimental acute T. cruzi infection is effective at reducing cardiac inflammation, fibrosis and conduction deficits. However, since the greatest burden of cardiac disease in patients with Chagas disease is in chronically infected patients, this manuscript would benefit from a discussion of how VEGF may play a role in cardiac disease in chronic infection.
Author Response
Dear Reviewer,
Thank you for sending the comments with recommended revision to the first version of our manuscript entitled “Vascular Growth Factor Inhibition with Bevacizumab Improves Cardiac Electrical Alterations and Fibrosis in Experimental Acute Chagas Disease”. According to the main comments, we made substantial changes in the manuscript and answered all comments, as you can find in detail below.
Thank you for the careful review process that improved our manuscript.
Best regards,
Roberto Ferreira
Reviewer 1
Overall, this is a well written manuscript identifying a role for VEGF in pathogenesis of cardiac inflammation and fibrosis in a mouse model of acute Trypanosoma cruzi infection, as a model of Chagas disease. The results have implications for developing novel therapies to improve treatment of Chagas disease. A few minor items should be clarified and the discussion should include implications for such a treatment in chronic disease.
- In the materials and methods, please list the source of the Male Swiss Webster mice. Were they bred in house or purchased from a commercial vendor?
Thank you for the question. The male Swiss Webster mice were bred in-house in the Institute of Science and Technology in Biomodels (ICTB) from Oswaldo Cruz Foundation which provides the animals for the laboratory.
- What was the source of the T. cruzi Y parasites used for infection? Were they maintained in mammalian cell culture or obtained as blood form trypomastigotes from infected mice?
The blood forms trypomastigotes were maintained through Swiss Webster mice infections, providing the parasites for the experimental infections.
- In line 111, please remove “de”
Thank you for the comment, “de” was removed in line 111.
- In section 2.1 or 2.2, please specify how many days after infection and/or treatment mice were euthanized for histologic analysis.
The sentence: “euthanized at 8 and 15 day post-infection (dpi)” was added in line 89 in section 2.1 and in line 159 in section 2.7.
- In line 121, please list the type of microscope and magnification used to analyze sections of heart stained with picrosirius red and hematoxylin/eosin.
Thank you for the comment, the sentence “3–5 animals samples were analyzed for each experimental group in Zeiss Axioplan 2 microscope at an overall magnification of 400” in lines 122-123.
- In line 304, please capitalize Chagas disease
Thank you for the comment, the correction was made in line 305.
- The data shown is very convincing that inhibition of VEGF in an experimental acute T. cruzi infection is effective at reducing cardiac inflammation, fibrosis and conduction deficits. However, since the greatest burden of cardiac disease in patients with Chagas disease is in chronically infected patients, this manuscript would benefit from a discussion of how VEGF may play a role in cardiac disease in chronic infection.
Thank you for the comment, the sentence: “Considering that a major problem with Chagas disease is chronic cardiomyopathy, in which there is well-established fibrosis, the VEGF blockage could be beneficial to minimize this aspect, however, further studies are necessary to understand the VEGF role at this stage” was added in lines 366-369 of the discussion.

Reviewer 2 Report
The authors investigated the effect of the VEGF inhibition with bevacizumab in an experimental murine model of acute Chagas disease. They evaluated the cardiac parasitism, inflammation, blood vessel abundance, heart fibrosis and Electrocardiographic alterations. They found that bevacizumab significantly increased survival, reduced inflammation, improved cardiac electrical function, diminished angiogenesis, decreased myofibroblasts in cardiac tissue and restored collagen levels. Overall, this manuscript is well organized, and shows very impressive conclusions.
However, my major concern is about the cardiovascular toxicity of bevacizumab. Because there are a large number of literature show that treatment with bevacizumab increases cardiovascular complications during the cancer therapy, such as hypertension, cardiac ischemia, and congestive heart failure. But this study shows that bevacizumab treatment seems have the protective effect to the heart and increased survival in treated T. cruzi-infected animals. What are the different mechanisms underlying those two different conditions? Can you give some more explanation in discussion?
Minor concerns:
P1L28-29: “Swiss Webster mice were infected with Y strain, clinical, morphological and molecular analyses were performed”. Please reorganize this sentence.
P3L110-112: What is the meaning of “de” in the sentence of “The parameters evaluated by de the software was heart rate….”
P7L198-201: A little confused by the sentence of “ At 15 dpi non-treated group presented an increase of 2.2-fold change …to infected non-treated group (p<0,01)”. Please re-structure and check the number including the fold change and p-value.
P8L224-226: Not sure how to calculate the percent of the reduced area as 23%. Please index the figure panels.
Author Response
Dear Reviewer,
Thank you for sending the comments with recommended revision to the first version of our manuscript entitled “Vascular Growth Factor Inhibition with Bevacizumab Improves Cardiac Electrical Alterations and Fibrosis in Experimental Acute Chagas Disease”. According to the main comments, we made substantial changes in the manuscript and answered all comments, as you can find in detail below.
Thank you for the careful review process that improved our manuscript.
Best regards,
Roberto Ferreira
Reviewer 2
The authors investigated the effect of the VEGF inhibition with bevacizumab in an experimental murine model of acute Chagas disease. They evaluated the cardiac parasitism, inflammation, blood vessel abundance, heart fibrosis and Electrocardiographic alterations. They found that bevacizumab significantly increased survival, reduced inflammation, improved cardiac electrical function, diminished angiogenesis, decreased myofibroblasts in cardiac tissue and restored collagen levels. Overall, this manuscript is well organized, and shows very impressive conclusions.
However, my major concern is about the cardiovascular toxicity of bevacizumab. Because there are a large number of literature show that treatment with bevacizumab increases cardiovascular complications during the cancer therapy, such as hypertension, cardiac ischemia, and congestive heart failure. But this study shows that bevacizumab treatment seems have the protective effect to the heart and increased survival in treated T. cruzi-infected animals. What are the different mechanisms underlying those two different conditions? Can you give some more explanation in discussion?
Thank you for the question, the sentence “Contradictory to our data, it was described higher cardiovascular risk factor in bevacizumab-treated cancer patients [54, 55], the mechanisms are not fully understood but may be the result of exacerbated inhibition of VEGF, decreasing nitric oxide NO and/or increasing proinflammatory gene expression which induces vasoconstriction and platelet aggregation. However, these different effects could be explained since those side effects were correlated to bevacizumab dose [54, 55] and close treatment monitoring should be carried out” was added in lines 359-365.
- Economopoulou P, Kotsakis A, Kapiris I, Kentepozidis N. Cancer therapy and cardiovascular risk: focus on bevacizumab. Cancer Manag Res. 2015 Jun 3;7:133-43. doi: 10.2147/CMAR.S77400. PMID: 26082660; PMCID: PMC4461138.
- Totzeck M, Mincu RI, Rassaf T. Cardiovascular Adverse Events in Patients With Cancer Treated With Bevacizumab: A Meta-Analysis of More Than 20 000 Patients. J Am Heart Assoc. 2017 Aug 10;6(8):e006278. doi: 10.1161/JAHA.117.006278. PMID: 28862931; PMCID: PMC5586462.
Minor concerns:
P1L28-29: “Swiss Webster mice were infected with Y strain, clinical, morphological and molecular analyses were performed”. Please reorganize this sentence.
Thank you for the comment, the sentence was rewritten “Swiss Webster mice were infected with Y strain and cardiac morphological and molecular analyses were performed” in lines 28-29.
P3L110-112: What is the meaning of “de” in the sentence of “The parameters evaluated by de the software was heart rate….”
The sentence was corrected, “de” was removed in lines 110-111.
P7L198-201: A little confused by the sentence of “ At 15 dpi non-treated group presented an increase of 2.2-fold change …to infected non-treated group (p<0,01)”. Please re-structure and check the number including the fold change and p-value.
Thank you for the comment, the sentence was rewritten: “Next, we determined the expression of VEGF in cardiac tissue by immunoblotting (Figure 3). At 15 dpi non-treated group presented an increase of 2.2-fold change in cardiac VEGF expression when compared to non-infected controls (p<0,01) and we did not observe changes in VEGF levels at 8 dpi when compared to non-infected controls (Figure 3A and 3B). The treatment with bevacizumab did not change VEGF levels at 8 dpi, however, at 15 dpi treatment with bevacizumab induced a reduction of 3.6-fold change in VEGF expression compared to the infected non-treated group (p<0,001) (Figure 3C and 3D)” in lines 197-203.
P8L224-226: Not sure how to calculate the percent of the reduced area as 23%. Please index the figure panels.
The figure information was added in the sentence “We observed that the treatment with bevacizumab - reduced (23%) the lectin-stained area (green in Figure 4E and 4F)” in lines 225-226.
